# A Long Contiguous Stretch of Homozygosity Disclosed a Novel *STAG3* Biallelic Pathogenic Variant Causing Primary Ovarian Insufficiency: A Case Report and Review of the Literature

**DOI:** 10.3390/genes12111709

**Published:** 2021-10-27

**Authors:** Simona Mellone, Marco Zavattaro, Denise Vurchio, Sara Ronzani, Marina Caputo, Ilaria Leone, Flavia Prodam, Mara Giordano

**Affiliations:** 1Laboratory of Genetics, SCDU Biochimica Clinica, Ospedale Maggiore della Carità, 28100 Novara, Italy; simona.mellone@uniupo.it; 2Endocrinology, Department of Translational Medicine, University of Eastern Piedmont, 13100 Novara, Italy; marco.zavattaro@med.uniupo.it (M.Z.); marina.caputo@uniupo.it (M.C.); 10033018@studenti.uniupo.it (I.L.); flavia.prodam@med.uniupo.it (F.P.); 3Laboratory of Genetics, Department of Health Sciences, University of Eastern Piedmont, 13100 Novara, Italy; denise.vurchio@uniupo.it (D.V.); sara.ronzani@uniupo.it (S.R.)

**Keywords:** *STAG3*, primary ovarian insufficiency, exome sequencing, LCSH

## Abstract

Primary ovarian insufficiency (POI) refers to an etiologically heterogeneous disorder characterized by hypergonadotropic hypogonadism that represents a major cause of infertility in women under 40 years of age. Most cases are apparently sporadic, but about 10–15% have an affected first-degree relative, indicating a genetic etiology. Pathogenic variations in genes involved in development, meiosis and hormonal signaling have been detected in the hereditary form of the disorder. However, most cases of POI remain unsolved even after exhaustive investigation. A 19-year-old Senegalese female affected by non-syndromic POI presented with primary amenorrhoea and answered well to the hormonal induction of puberty. In order to investigate the presence of a genetic defect, aCGH-SNP analysis was performed. A 13.5 Mb long contiguous stretch of homozygosity (LCSH) was identified on chromosome 7q21.13-q22.1 where the exome sequencing revealed a novel homozygous 4-bp deletion (c.3381_3384delAGAA) in *STAG3*. Pathogenic variants in this gene, encoding for a meiosis-specific protein, have been previously reported as the cause of POI in only eight families and recently as the cause of infertility in a male. The here-identified mutation leads to the truncation of the last 55 amino acids, confirming the important role in meiosis of the *STAG3* C-terminal domain.

## 1. Introduction

Primary ovarian insufficiency (POI) is characterized by depletion of ovary follicles, leading to hypoestrogenism and hypergonadotropic hypogonadism, infertility and amenorrhea in women younger than 40 years [1,2]. This disorder represents one of the main causes of infertility affecting women, with a worldwide prevalence of approximately 1–3.7% [3,4,5]. Women with POI may show a wide range of clinical phenotypes, from primary (PA) to secondary amenorrhea (SA) and other congenital or acquired abnormalities, and present either syndromic or isolated forms of the disease [6,7]. PA is usually diagnosed in adolescence in patients showing delayed puberty and absence of secondary sex characteristic development, while SA, which represents the most frequent POI phenotype, may occur at any age after menarche and is characterized by normal pubertal development [8,9]. Although POI is a heterogeneous disorder caused by iatrogenic, viral or autoimmune factors, more than 70% of cases remain idiopathic [1]. Chromosomal aberrations have long been recognized as a cause of POI and at least 10–13% of the syndromic cases present anomalies revealed by standard karyotype [10]. About 10–15% of women with POI have an affected first-degree relative and different modes of inheritance can be observed in families, suggesting the presence of several monogenic causes in the etiology of the disease [7]. To date, the use of different genomic approaches, including linkage studies, sequencing of candidate genes and whole exome sequencing (WES) allowed the identification of pathogenic alterations in more than 60 genes implicated in both syndromic and isolated POI [1,2,3,4,5,6,7,8,9,10]. Additionally, submicroscopic copy number variations (CNVs) encompassing genes potentially implicated in reproductive function have emerged as an important genetic determinant in POI [11]. The genes altered in POI encode for protein involved in DNA repair and meiosis [1] and pathogenic variants predispose to different forms of cancer [6,12,13,14]. Thus, the early diagnosis of the molecular mechanism underlying POI is essential to develop strategies for preventing the irreversible consequences on fertility, to improve clinical management and to perform genetic counseling with a long-term follow-up also considering tumor susceptibility.

Pathogenic variants in *STAG3* (stromal antigen 3), which encodes a subunit of the cohesin complex participating to sister chromatid pairing during meiosis, have been identified as a rare POI monogenic cause. To date *STAG3* biallelic variants have been reported in eight families worldwide, five of which were consanguineous pedigree [6,8,13,15,16,17,18,19]. All the affected women had isolated POI except for a patient belonging to a Palestinian family [6] who presented simultaneous bilateral ovarian tumors.

Here we report the identification of a novel biallelic pathogenic variation in *STAG3* through a combined approach of CGH-SNP microarray and Clinical Exome Sequencing (CES) in a young Senegalese woman affected by non-syndromic POI.

## 2. Materials and Methods

### 2.1. Case Presentation

The patient was a 19-year-old female from Senegal, referred to the Emergency Room Department for dyspareunia and pelvic pain after her first sexual intercourse. She was also suffering from primary amenorrhea and anosmia. Familiar anamnesis revealed the absence of parental consanguinity. Computerized tomography (CT) investigation showed vaginal stenosis and failure to display ovaries. Physical examination showed absence of breast and pubic hair development, consistent with Tanner stage 1, although stature was normal for age. Hormonal profile was indicative of primary hypogonadism (17ß-estradiol 20 pg/mL, FSH 88.1 mU/mL, LH 28.7 mU/mL), while the remaining pituitary function was preserved (TSH 1.4 µU/mL, free-T_4_ 1.33 ng/dL, prolactin 9.78 ng/mL, GH 1.99 ng/mL, IGF-I 271.8 ng/mL, ACTH 22.4 pg/mL, 8.00 a.m. cortisol 11.7 µg/dL). Further investigations revealed reduced bone mineral density at lumbar spine (Z-score −3.6) and delayed hand bone age (14 years). Moreover, pelvic magnetic resonance (MR) confirmed the presence of a pre-pubertal uterus, characterized by hypoplasia (38 × 11 × 19 mm) and a fundus-to-cervix ratio of 1:1 (Figure 1, panel a,b). The cytogenetic analysis revealed a normal karyotype (46,XX).

After hormonal and radiological diagnostic work-up, a puberty induction therapy using transdermal 17ß-oestradiol (estradiol hemyhidrate 25 µg/3 days for 2 months, then 50 µg/3 days) was started. After 3 months of treatment, the presence of pubic hair (P2) and breast development (B2), consistent with Tanner stage 2, were appreciable at physical examination. Six months since the beginning of puberty induction therapy, breast size and pubic hair further increased (B3 P3) and menarche occurred. Hormonal evaluation showed adequate 17ß-oestradiol levels (75 pg/mL) and consensual reduction in gonadotropin concentration (FSH 4.7 mU/mL, LH 0.4 mU/mL). Treatment with transdermal estrogen led to oligomenorrhea and progressive pubertal development over the following months. After 24 months of therapy, complete pubertal (B5 P4) and uterine development occurred, and combined therapy with progesterone was started (Table 1). Pelvic magnetic resonance performed at that time revealed an anteverted-antiflex uterus of normal size and regular vaginal morphology; endometrial thickness was normal according to menstrual phase (Figure 1, panel c,d).

### 2.2. CGH-SNP Microarray Analysis

Genomic DNA of the proband was extracted from peripheral blood through the ReliaPrep Blood gDNA Miniprep System (Promega), according to the manufacturer’s recommendation. Medium-density microarray analysis was performed using standard protocols with a GenetiSure Dx Postnatal Array 4 × 180K + SNP (Agilent Technologies, Santa Clara County, CA, USA). The SNP array contains 59,000 single-nucleotide polymorphism probes and 107,000 oligonucleotide probes (60-mer) with a resolution of 5~10 Mb for ROH (Region of Homozygosity) detection. The slides were scanned through Agilent SureScan Dx Microarray Scanner System (Agilent Technologies, USA). Image analysis, normalization and annotation were based on Agilent Feature Extraction Software. Finally, CNVs were called with Agilent CytoDx Software 1.1.1.0 using tiff images from data.

### 2.3. Clinical Exome Sequencing(CES)

The Agilent SureSelect Custom Constitutional Panel 17 Mb (Agilent, Santa Clara, CA, USA) that includes 5227 clinically relevant genes was used for the preparation of the library. This custom panel also covers all 65 genes from the Genomics England Panel App for primary ovarian insufficiency (https://panelapp.genomicsengland.co.uk/ accessed on 26 October 2021). Exon-enriched library was subjected to a 150 bp paired-end sequencing on the Illumina MiSeq platform (Illumina, San Diego, CA, USA). Sequencing reads passing quality filters were aligned to the human reference genome build (GRGh37/hg19) and variant calling was performed using the SureCall v3.5 software (Agilent Technologies). Then VCF files were annotated with the wANNOVAR tool. After that, the variants were filtered and prioritized using a personalized bioinformatics pipeline: variants with a read coverage of less than 5× and a Qscore of below 20 were filtered out; an allele frequency in public databases <0.1% (1000 Genomes, the ESP cohort data set, GnomAD, Exome Aggregation Consortium) was considered; synonymous variants were excluded. For the missense variants, we used at least four in silico prediction tools (SIFT, CADD, Polyphen, MutationTaster) through the Varsome website (https://varsome.com/ accessed on 26 October 2021) [20].

### 2.4. Sanger Sequencing

The *STAG3* variant (NM_001282716.1:c.3381_3384delAGAA) (p.Glu1128Metfs*42) identified by clinical-exome sequencing was validated by Sanger sequencing using the following primers designed by the Primer3 (v. 0.4.0) software: forward: 5′-TTGGAAAGAGAGCACACCTG-3′ and reverse: 5′-TGGTGTTAATGGGGAGAAAA-3′.

## 3. Results

An array CGH-SNP assay was used as first tier test to investigate the presence of both DNA copy number variations (CNVs) and loss of heterozygosity (LOH) regions. No chromosomal unbalance was identified. However, the aCGH-SNP revealed a LCSH (long contiguous stretches of homozygosity) region of 13.5 Mb (from rs2374083 to rs1990167) on chromosome 7q21.13-q22.1. A LCSH refers to a disomic long homozygous stretch and is used here instead of LOH as the latter describes an event where heterozygosity was originally present and now is lost. This region contained 127 OMIM genes (Figure 2A) among which the best candidate to explain the patient’s phenotype was *STAG3*, whose biallelic mutations had been previously detected in patients with autosomal recessive POI [6,8,13,15,16,17,18,19].

Due to the large size of *STAG3* (34 exons) and to eventually disclose pathogenic variants in other genes a NGS platform including 5227 genes (Agilent SureSelect Custom Constitutional Panel 17) involved in human genetic disorders (Clinical Exome) was used to identify the causative molecular defect. The average sequencing depth was 77x with 96.93% of the target sequence covered at ≥20×, 89.27% covered at ≥50× and 70.52% covered at ≥100×. The variants obtained from CES were filtered as previously described and confirmed the presence of a LCSH on chromosome 7q21.13-q22.1. A variant at the homozygous state in exon 30 of the *STAG3* gene was identified, namely c.3381_3384delAGAA (NM_001282716.1; Figure 2B), then validated by Sanger sequencing. This 4-bp deletion (NC_000007.13:g.99808776_99808779del) causes a frameshift with the introduction of a premature stop codon (p.Glu1128Metfs*42) resulting in a predicted truncated protein of 1170 amino acids (wild-type *STAG3* protein: 1225 amino acids). The variant was not present in the public databases and was classified as pathogenic according to the American College of Medical Genetics and Genomics (ACMG) standards and guidelines for the interpretation of variations (criteria: PVS1, PM2, PP3) [21]. We did not find any other pathogenic or likely pathogenic homozygous variant in genes involved in POI or in other that passed through the set filters (<0.1% in population databases, >20 Q score and >5× coverage), consequently *STAG3* (p.Glu1128Metfs*42) was considered the best disease-causing candidate variant in this patient. Sanger sequencing confirmed the homozygous variant in the patient’s DNA. Unfortunately, the parents’ DNA was not available to investigate the origin of this LCSH.

## 4. Discussion and Conclusions

POI is an extreme heterogeneous disorder caused by pathogenic variations in genes involved in germ cell development, oogenesis, folliculogenesis, steroidogenesis and hormone signaling [22]. Chromosomal unbalances detected through standard karyotype as well as submicroscopic alterations are involved in syndromic forms of POI [10]. Ledig et al. [23] reported the identification of several micro-rearrangements detected through aCGH in 74 German POI patients, encompassing several genes involved in meiosis, DNA repair and folliculogenesis.

In the present patient a platform combining aCGH and SNP array was used as the first-tier genetic test. In addition to CNVs, this platform detects the presence of homozygosity regions through SNP genotyping as the consequence of (i) hemizygosity caused by a CNV, that can be simultaneously confirmed by CGH probes; (ii) autozygosity of identity-by-descent regions underlying the presence of a putative recessive disorder; and (iii) uniparental disomy (UPD) when the homozygosity is confined to a chromosomal region (or to entire chromosome) underlying an imprinting disorder. In the patient described here, the CGH-SNP array revealed a normal female karyotype and the presence of a unique interstitial long contiguous stretches of homozygosity region (LCSH) of 13.5 Mb on chromosome 7q21.13-q22.1. Besides the 7q LCSH, no other large regions (>5 Mb) of homozygosity were detected, indicating either segmental UPD [24] or a distant shared parental ancestry. The origin of this LCSH might have been explored through the parents’ DNA, which was not available as both father and mother currently live in Senegal.

The best candidate in the LCSH region was *STAG3*, which has been previously associated with a recessive form of POI consistent with the patient’s phenotype. *STAG3* encodes a subunit of the multiprotein cohesin complex required during meiosis I for homologous chromosome pairing, correct synapsis and segregation of chromosomes, proper recombination and DNA repair [25]. In mammals, meiosis-specific cohesin subunits include a SMC1 subunit (SMC1β), two additional α-kleisins (RAD21L and REC8) and a stromal antigen protein (*STAG3*) that are specifically expressed in meiosis. Moreover, Stag3 specifically localizes on the spindle apparatus and is required for microtubule stability and spindle assembly to preserve the euploidy in the mouse eggs [26].

Females *Stag3^−/−^* mice lack ovarian follicles indicating a severe ovarian dysgenesis [27] and also develop ovarian tumors as observed in humans [6]. It is likely that homozygous or compound heterozygous variants in *STAG3* may also affect male fertility due to the early prophase I arrest and apoptosis in both sex germ cells. Indeed, all *Stag3^−/−^* male mice described to date are infertile [28,29] and a biallelic loss-of-function *STAG3* variant has been recently reported in infertile males [28,29,30] affected by complete bilateral meiotic arrest.

Up to date, only 11 variants in *STAG3* have been reported as genetic causes of POI in eight pedigrees (Figure 3) [6,8,13,15,16,17,18,19] with a predicted loss of function effect in most cases. The *STAG3* frameshift variant identified here is a 4-bp deletion in exon 30 and the resulting transcript would either undergo nonsense-mediated decay [31] or result in a truncated protein devoid of the C-terminal domain similarly to a previously described frameshift mutation truncating the last 206 C-terminal residues in exon 28 [13]. The C-terminal domain is specific of *STAG3* as it is not conserved neither in *STAG1* nor in *STAG2*, which belong to the same family but are involved in somatic cell division [32]. As STAG3 is specifically expressed during meiosis [6,13] the effect of the variant identified here on protein/mRNA level could not be performed. However as the variant does not fall in the last exon of the transcript, it can be hypothesized that RNA nonsense mediated decay may occur in vivo.

The finding of two POI patients with pathogenic variants truncating the C-terminal can support an important specific role of this domain of *STAG3* during meiosis.

Our patient presented with primary amenorrhea, absence of breast development, hypoplasia of the uterus, osteoporosis and normal height. These clinical features were also reported in other individuals carrying *STAG3* pathogenic variants [6,8,13,15,16,17,18,19]. The hormone replacement therapy (HRT) allowed for puberty to be induced and for the patient to achieve complete secondary sexual characteristics, adequate growth, uterine development and menstrual fluxes. HRT currently represent the mainstay therapy for women with POI and plays a pivotal role in reducing long-term comorbidities (e.g., osteoporosis and cardiovascular disease) and improving sexual health by restoring normal serum estrogen concentrations according to age [3,9,33].

Taken together, our observations strength the importance of a correct diagnosis and clinical management in women with POI. The diagnostic workflow should include the molecular test with multi-gene panel including *STAG3* to allow an appropriate ovarian monitoring in the long-term follow-up, considering the high risk to develop ovarian tumors.

## Figures and Tables

**Figure 1 genes-12-01709-f001:**
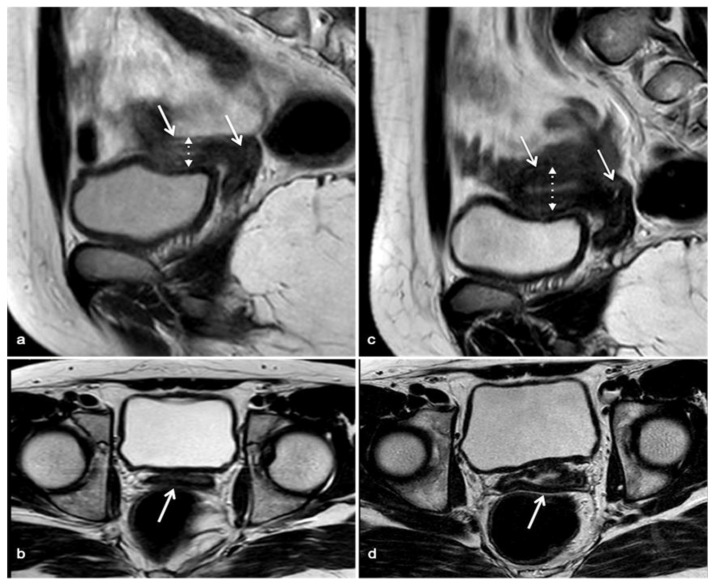
Magnetic resonance before and after estrogen replacement therapy. Panels “**a**”,“**b**” show uterine hypoplasia (continuous white arrows) in sagittal (panel “**a**”) and transversal plane (panel “**b**”). Reduced uterine thickness, consistent with pre-pubertal morphology, is evidenced in panel “**a**” (dashed arrow). In panels “**c**”,“**d**”, continuous white arrows indicate the uterine development after 24 months of estrogen therapy both in sagittal (panel “**c**”) and transversal (panel “**d**”) planes. Endometrial development and increase in uterine thickness are evident in panel “**c**” (dashed arrow).

**Figure 2 genes-12-01709-f002:**
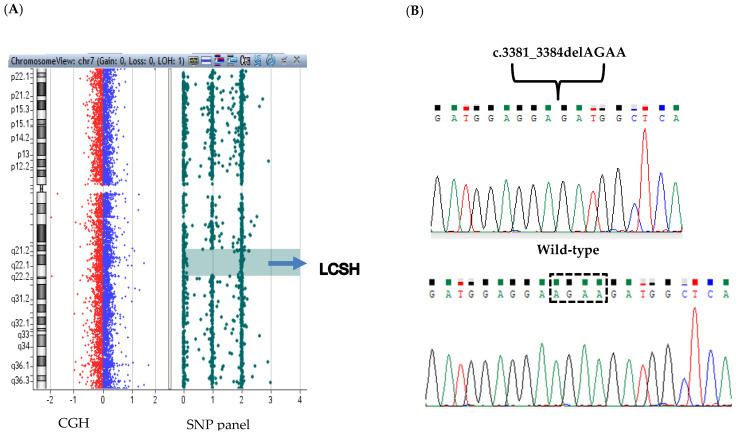
CGH-SNP array and sequencing results. (**A**) Array-CGH-SNP results: the LCSH on chromosome 7q21.13-q22.1 is indicated by a blue arrow. (**B**) The partial sequencing electropherogram is shown including the novel homozygous mutation identified in STAG3 (NM_001282716.1:c.3381_3384del); the corresponding wild-type sequence is shown below with a dashed rectangle showing the deleted bases.

**Figure 3 genes-12-01709-f003:**
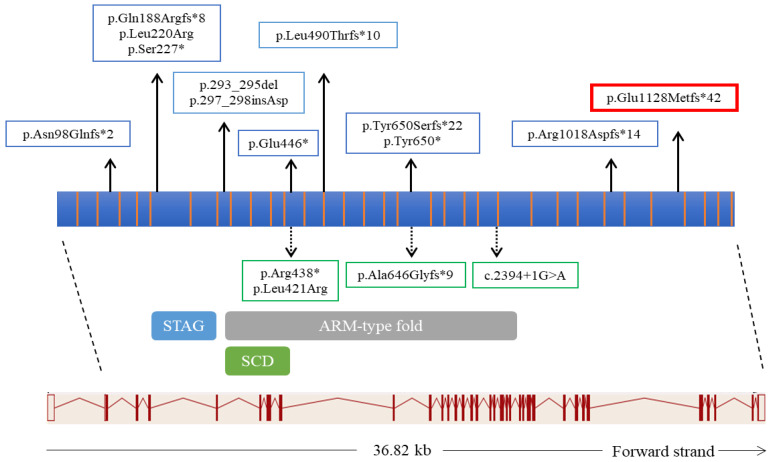
Schematic representation of *STAG3* and protein structure with pathogenic variants reported in POI patients. The exon-intron structure of *STAG3* (transcript NM_001282716.1) is displayed in the lower part of the figure. The scheme of STAG3 protein consisting of 1225 amino acids (aa) (protein domains for ENSP00000400359.1) is reported in the middle part of the figure with colored rectangles indicating the STAG domain (174–283 aa; PF08514 Pfam database), an Armadillo-type fold domain (ARM-type fold) (303–813 aa; SSF48371 Superfamily database) and a Stromalin conservative domain (SCD) (309–394 aa; PS51425 Prosite profiles). The pathogenetic variants identified in previous studies in females are shown in blue boxes whereas the variant identified in this study is shown in the red box. The pathogenetic variants identified in males are shown in green boxes.

**Table 1 genes-12-01709-t001:** Hormonal and clinical changes over time of the described clinical case. B: breast; P: pubic hair; FSH: Follicular-Stimulating Hormone; LH: Luteinising Hormone.

	November 2018 (Staring Therapy)	February 2019 (+3 Months)	May 2019 (+9 Months)	September 2019 (+10 Months)	June 2020 (+18 Months)	November 2020 (+24 Months)
Tanner stage (BP)	Stage 1 (B1,P1)	Stage 2 (B2,P2)	Stage 3 (B3,P2-3)	Stage 4 (B4,P3)	Stage 4 (B4,P4)	Stage 5 (B5,P4)
17 β-oestradial(pg/mL)	20	-	75	-	27	40
FSH(mU/mL)	88.1	-	4.7	-	55.6	51.1
LH(mU/mL)	28.7	-	0.4	-	24.2	29.9
Menstrual cycle	Amenorrhea	Amenorrhea	Menarche	Oligomenorrhea	Euhenorrhea	Oligomenorrhea
Therapy -oestradial hemyhidrate -progesterone	25 μg/3 days	50 μg/3 days	50 μg/3 days	50 μg/3 days	50 μg/3 days	50 μg/3 days 100 mg/14 days

## Data Availability

Not applicable.

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
