# Peer review of "A Long Contiguous Stretch of Homozygosity Disclosed a Novel STAG3 Biallelic Pathogenic Variant Causing Primary Ovarian Insufficiency: A Case Report and Review of the Literature"

_genes, 2021, doi:10.3390/genes12111709_

Round 1
Reviewer 1 Report
The paper describes a novel truncating STAG3 variant in a POI patient with a review of STAG3 literature. A frameshifting STAG3 variant was identified using a combination of array CGH which pinpointed to a long contiguous stretch of homozygosity on chromosome 7 where STAG3 resides. A customised exome panel with 5227 genes identified a 4-bp deletion in STAG3. STAG3 variants have been previously described but are a rare cause of POI.
The case is well described and contributes towards the understanding of STAG3 variation leading to the POI phenotype, but the paper lacks attention to detail with a few typos, nomenclature inconsistencies and missing literature. Lack of parental DNAs and mRNA work which would demonstrate whether the truncated mRNA is subjected to nonsense-mediated decay or resulting in a truncated protein is a shortcoming of the study. In my opinion, this case report with literature review is acceptable for publication providing the points raised below are addressed and improved/corrected.
Nomenclature comments
Please check HGVS nomenclature guidelines at https://varnomen.hgvs.org/ and make sure gene symbols are in italics throughout the paper when referring to the DNA or RNA (eg STAG3).
A note of caution to the authors - a couple of previous STAG3 publications had erroneous variant nomenclature (Caburet et al 2014 and also Riera-Escamilla A et al 2019). Riera-Escamilla et al published an erratum. Please pay attention to this when describing previously described STAG3 variants as part of the literature review. Caburet et al 2014 paper misinterpreted their variant as p.F187fs*7. If you check their sequence, the variant is actually chr7:g.99786486del causing p.(Gln188Argfs*8).]
Could you please also submit your variant to either ClinVar or LOVD database?
Specific nomenclature comments:
- Please remove redundant reference bases from the variant description. This is in line with HGVS nomenclature guidelines (you can use VariantValidator or Mutalyzer). Correct description on cDNA level is: NM_001282716.1:c.3381_3384del. Alternatively, you could use MANE SELECT transcript (coordinates are still the same): NM_001282717.1:c.3381_3384del https://www.ncbi.nlm.nih.gov/refseq/MANE/
- Protein level: predicted consequences, i.e. without experimental evidence (no RNA or protein sequence analysed), should be given in parentheses, ie p.(Glu1128MetfsTer43). Variant validator predicts 1:c.3381_3384del leads to p.(Glu1128MetfsTer43); the authors interpreted it as p.(Glu1128MetfsTer42). Please check with Mutalyzer which one is correct.
- Could you please also include genomics coordinates? For example: NC_000007.13:g.99808776_99808779del (see HGVS nomenclature guidelines)
- Page 5, row 182: you previously used 3-letter amino acid codes; please use 3-letter codes throughout the manuscript
- Please consistently use either '*' or 'Ter' when describing truncating changes. Both are OK, but please stick to one.
Introduction
Page 1 row 24: review sentence please; “well answered to … “; should it have been “answered well to…”?
Page 1 row 31: typo; amino acids (two words)
Page 1, row 38/39: Please check the sentence about the prevalence of POI. It is confusing whether the prevalence of POI is 3.7% or 1%. Please check references.
Page 2, row 63: The use of the word ‘mutation’ is discouraged by the HGVS guidelines; please check HGVS nomenclature guidelines and use the work ‘variant’ (pathogenic) instead of mutation.
Page 2, row 65/66: There is another (very recent) publication describing a biallelic loss of function variants (nonsense STAG3 variant and whole gene deletion). Could you please add: Demain LAM, Boetje E, Edgerley JJ, Miles E, Fitzgerald CT, Busby G, Beaman GM, O'Sullivan J, O'Keefe RT, Newman WG. Biallelic loss of function variants in STAG3 result in primary ovarian insufficiency. Reprod Biomed Online. 2021 Jul 16:S1472-6483(21)00343-6. PMID: 34497033.
Materials and Methods
Figure 1 (legend) says ‘continuous white arrows’; there are only black arrows in the Figure. The authors also mention a dashed arrow – I can’t see a dashed arrow in Figure 1? Legend can be made easier to read: a. Uterine hypoplasia sagittal plane (solid arrow); reduced uterine thickness consistent with pre-pubertal morphology (dashed arrow) b. Uterine hypoplasia transversal plane (solid arrow) … Please make the Legend consistent with the figure.
Page 4, row 135: ROH abbreviation needs to be written out the first time it is used; ie Region of Homozygosity (ROH)
Page 4, row 152: close bracket missing after ExAC
Page 4, row 154: typo: through, not ‘throw’
Page 4, row 154: Please add citation to Varsome; Varsome requirement: “If you use VarSome for your work, please cite our paper in your publications and other communications: VarSome: the human genomic variant search engine, Christos Kopanos, Vasilis Tsiolkas, Alexandros Kouris, Charles E Chapple, Monica Albarca Aguilera, Richard Meyer, Andreas Massouras, Bioinformatics, Volume 35, Issue 11, 1 June 2019, Pages 1978–1980.”
Clinical Exome sequencing: Could the authors please confirm that all genes relevant for POI were on the custom panel of 5227 clinically relevant genes? For example – are all the genes from the Genomics England Panel App for Primary ovarian insufficiency included (65 genes)
Results:
- Is there a family history of POI? No additional affected or unaffected 1st or 2nd-degree relatives reported? The authors do comment (in the discussion) that parental DNAs are not available.
- Were there any other homozygous variants that passed through the set filters (<0.1% in population databases, >20 Q score and >5x coverage)?
Discussion
- It would be helpful if the authors added a paragraph discussing how the clinical features of their POI patient relate to other published patients with STAG3 pathogenic variants.
- Could the authors please point out the lack of RNA and protein work that would clarify the effect of the deletion on protein/mRNA level is a shortcoming of this case report?
- Page 6, row 257: I think there is possibly another paper describing STAG3 and azoospermia in men; can you please check this paper: Riera-Escamilla A, et al. Sequencing of a 'mouse azoospermia' gene panel in azoospermic men: identification of RNF212 and STAG3 mutations as novel genetic causes of meiotic arrest. Hum Reprod. 2019 Jun 4;34(6):978-988. Please pay attention - the authors of the paper had a correction published: “The variant NM_001282718:c.1759dupG should be NM_001282718.2:c.1762dupG:p.(Ala588GlyfsTer9); referring to the largest transcript the variant is equivalent to NM_012447.3:c.1936dupG:p.(Ala646GlyfsTer9).”
Figure 3 comments:
- There is a “p.” missing in "Ser227*"
- When you describe your STAG3 variant n Methods, you used transcript: NM_001282716.1. In Figure 3 you use the Ensembl ENST00000426455.5. For consistency, could you please use one transcript and make sure all previously described changes are interpreted according to the same transcript?
- Could you please add STAG3 variants from the recent paper I mentioned before (Demain LAM at al 2021; PMID: 34497033).
- Could you please consider adding the few STAG3 variants which were described as the cause of male infertility?
Reviewer 2 Report
The manuscript is well-written, which delivered the main idea clearly. It will be better to cover the limitations of the study at length.
